# Development and Clinical Application of a Molecular Assay for Four Common Porcine Enteroviruses

**DOI:** 10.3390/vetsci11070305

**Published:** 2024-07-09

**Authors:** Zhonghao Xin, Shiheng Li, Xiao Lu, Liping Liu, Yuehua Gao, Feng Hu, Kexiang Yu, Xiuli Ma, Yufeng Li, Bing Huang, Jiaqiang Wu, Xiaozhen Guo

**Affiliations:** 1Key Laboratory of Poultry Disease Diagnosis and Immunity in Shandong Province, Poultry Research Institute, Shandong Academy of Agricultural Sciences, Jinan 250100, China; 18366592378@163.com (Z.X.); lishiheng1997@163.com (S.L.); luxi0515@163.com (X.L.); liuliping@saas.ac.cn (L.L.); gaoyuehua@saas.ac.cn (Y.G.); hufeng@saas.ac.cn (F.H.); yukexiang@saas.ac.cn (K.Y.); maxiuli@saas.ac.cn (X.M.); liyufeng@saas.ac.cn (Y.L.); huangbing@saas.ac.cn (B.H.); 2College of Veterinary Medicine, Inner Mongolia Agricultural University, Hohhot 010020, China; 3Shandong Key Laboratory of Disease Control and Breeding, Institute of Animal Science and Veterinary Medicine, Shandong Academy of Agricultural Science, Jinan 250100, China; wujiaqiang@saas.ac.cn

**Keywords:** TaqMan probe, multiplex real-time RT-PCR, porcine enteroviruses, diagnosis

## Abstract

**Simple Summary:**

Porcine epidemic diarrhea virus (PEDV), porcine transmissible gastroenteritis virus (TGEV), porcine deltacoronavirus (PDCoV), and porcine rotavirus-A (PoRVA) are the four significant pathogens of viral diarrhea of piglets in large-scale pig farms, which have caused huge economic losses to the pig industry all over the world. Since these four viruses have very similar clinical symptoms, it is necessary to develop an excellent detection method that can differentiate and diagnose these four viruses. In this study, we developed a multiplex real-time RT-PCR method that can simultaneously differentiate and diagnose these four viruses. The method has high specificity and does not cross-react with other porcine viruses. The lower limit of detection was 2.18 × 10^2^ copies/μL. In addition, we tested 97 clinical samples collected by this method, and the results were consistent with the detection results of traditional RT-PCR, indicating that this method is reliable. In summary, we developed a multiplex real-time RT-PCR method for simultaneous detection of PEDV, TGEV, PDCoV, and PoRVA, and the results of this study can provide technical means for the differential diagnosis and epidemiological investigation of these four porcine diarrhea virus diseases.

**Abstract:**

Porcine epidemic diarrhea virus (PEDV), porcine transmissible gastroenteritis virus (TGEV), porcine deltacoronavirus (PDCoV), and porcine rotavirus-A (PoRVA) are the four main pathogens that cause viral diarrhea in pigs, and they often occur in mixed infections, which are difficult to distinguish only according to clinical symptoms. Here, we developed a multiplex TaqMan-probe-based real-time RT-PCR method for the simultaneous detection of PEDV, TGEV, PDCoV, and PoRVA for the first time. The specific primers and probes were designed for the M protein gene of PEDV, N protein gene of TGEV, N protein gene of PDCoV, and VP7 protein gene of PoRVA, and corresponding recombinant plasmids were constructed. The method showed extreme specificity, high sensitivity, and excellent repeatability; the limit of detection (LOD) can reach as low as 2.18 × 10^2^ copies/μL in multiplex real-time RT-PCR assay. A total of 97 clinical samples were used to compare the results of the conventional reverse transcription PCR (RT-PCR) and this multiplex real-time RT-PCR for PEDV, TGEV, PDCoV, and PoRVA detection, and the results were 100% consistent. Subsequently, five randomly selected clinical samples that tested positive were sent for DNA sequencing verification, and the sequencing results showed consistency with the detection results of the conventional RT-PCR and our developed method in this study. In summary, this study developed a multiplex real-time RT-PCR method for simultaneous detection of PEDV, TGEV, PDCoV, and PoRVA, and the results of this study can provide technical means for the differential diagnosis and epidemiological investigation of these four porcine viral diarrheic diseases.

## 1. Introduction

Porcine enteric viruses pose significant challenges in pig farming worldwide. These viruses are responsible for causing acute diarrhea in piglets, leading to substantial economic losses in the swine industry worldwide [1], including porcine epidemic diarrhea virus (PEDV), porcine deltacoronavirus (PDCoV), porcine transmissible gastroenteritis virus (TGEV), porcine enteric alphacoronavirus (PEAV), porcine hemagglutinating encephalomyelitis virus (PEHV), porcine norovirus (PNoV), etc. Among these porcine enteroviruses, coronaviruses are a massive viral family that can cause digestive and respiratory tract diseases in humans and animals, posing a serious threat to human and animal health [2,3]. The prevalence of porcine enteric coronaviruses (PECs) affects scaled swine farms globally and creates a potential risk of cross-species transmission [4,5]. Of these, the three most vital and prevalent viruses are TGEV, PEDV, and PDCoV; previous research findings have shown that the sequencing findings of the swine enteric coronavirus (SeCoV) observed in various European countries, along with the Canine CoronaVirus-Human Pneumonia-2018 (CCoV-HuPn-2018) virus discovered in Malaysia, both indicate a strong connection between these recently discovered coronaviruses and TGEV [6]. Recently conducted research has indicated that human small intestinal epithelial cells can be infected by PEDV, thereby suggesting the likelihood of PEDV being transmitted across different species [7]. In the year 2021, scientists from the United States reported that blood samples taken from three Haitian children who were experiencing unexplainable fever tested positive for PDCoV. This finding suggests the potential of PDCoV transmission between different species [8,9]. Nevertheless, it is important to note that rotavirus (RV) is a major cause of acute gastroenteritis in animals, particularly nursing and weaned piglets, and should not be disregarded [10,11]. Ten categories (A to J) of RVs can be distinguished by the antigenic connections found in their VP6 proteins. Among these groups, A, B, and C (RVA, RVB, and RVC) are the most prevalent and infect both humans and animals. RVA strains, in particular, have been extensively studied and are considered one of the leading causes of acute dehydrating diarrhea from the perspectives of both public health and veterinary health. Numerous global studies have established a link between diarrheal episodes and RVC infections [12,13,14,15]. In addition, it is worth noting that the control effects have been insufficient in recent years due to the vaccine’s specificity and the prevalence of various RV genotypes or serotypes. Meanwhile, there have been reports of genetic reassortment between porcine rotavirus (PoRVA) and bovine/human rotavirus, indicating the existence of interspecies transmission between animals and humans [12,16,17].

The four viruses, PEDV, TGEV, PoRVA, and PDCoV, can lead to similar clinical symptoms and pathologic changes in pigs, which are difficult to distinguish [18]. In recent years, diarrhea caused by porcine enteric coronaviruses has occurred frequently in pig farms, and co-infections and secondary infections of PEDV, PoRVA, TGEV, and PDCoV have been common [18,19], which has made differential diagnosis difficult in clinical practice [19,20,21,22]. Consequently, it is crucial to develop a cost-effective, rapid, and accurate method for the detection and differential diagnosis of these viral swine diarrhea pathogens. So far, several relevant and good multiplex qPCR methods have been developed to address the above problems [23,24]. However, these methods were primarily designed for PECs alone, ignoring the increase in clinical cases of PoRVA with co-infections in swine herds and their current epidemiological significance. Here, we developed a TaqMan-probe-based multiplex real-time RT-PCR method for the simultaneous differential diagnosis of PEDV, TGEV, PDCoV, and PoRVA, which, to the best of our knowledge, is the first quadruplex RT-qPCR method for the simultaneous detection of the three major PECs and PoRVA. The development of this method can provide a technical tool for differential diagnosis, molecular epidemiological investigation, prevention, and control of swine diarrhea virus diseases.

## 2. Materials and Methods

### 2.1. Viruses and Clinical Samples

The virus samples were as follows: PEDV, TGEV, PDCoV, PoRVA, pseudorabies virus (PRV), porcine circovirus 2 (PCV2), porcine reproductive and respiratory syndrome virus (PRRSV), classical swine fever virus (CSFV), and Seneca Valley virus (SVV); these were previously preserved in our laboratory and confirmed by conventional PCR and genetic sequencing (Tsingke biotech, Qingdao, China).

Clinical test samples were obtained from 97 clinical diarrhea samples, which mainly included small intestines, intestinal contents, and feces, sent from large-scale in various regions of Shandong Province from 2020 to 2023. All samples were preserved at −80 °C in our laboratory.

### 2.2. Primers and Probes

The reference strains used to design primers and probes were derived from GenBank-logged gene sequences. Based on the virus strains, PEDV-HNAY 2016 (Accession No. MT338518.1), TGEV-KT2 (Accession No. JQ693059.1), PDCoV-USA/Ohio 137/2014 (Accession No. KJ601780.1), and PoRVA-DZ-2 (Accession No. KT820775.1) were designed to target the PEDV-M, TGEV-N, PDCoV-N, and PoRVA-VP7 genes and used to design primers and probes. Four pairs of primers and probes were designed on the NCBI Primer-BLAST website (Primer designing tool (nih.gov)) and synthesized by Tsingke Biotech (Qingdao) Co., Ltd. The information on primers and probes is shown in Table 1.

### 2.3. RNA Extraction and Reverse Transcription

All clinical samples were resuspended with 10 mL of saline solution, vortexed, and centrifuged at 12,000 rpm at 4 °C for 15 min. Nucleic acids were extracted using the MiniBEST Viral RNA/DNA Extraction Kit (Beijing Takara Biotech Co., Ltd., Beijing, China), following the manufacturer’s instruction. Reverse transcription was performed using the PrimeScript™ Ⅳ 1st strand cDNA Synthesis Mix (Beijing Takara Biotech Co., Ltd.).

### 2.4. Construction of Recombinant Plasmids for Standard Curves

The cDNAs of PEDV, TGEV, PDCoV, and PoRVA were used as templates, respectively, and the target gene fragments were amplified by PCR with the corresponding primers in Table 1. The amplified PCR products were recovered by a PCR purification kit, and the recombinant plasmids were identified by PCR and DNA sequencing after being ligated with the pMD18-T vector.

In order to establish a standard curve, we determined the concentration of the plasmids of the four viruses, calculated the copy number of the plasmids according to the formula, diluted the four plasmids 10-fold, and then diluted the plasmids from 2.18 × 10^8^ copies/μL to 2.18 × 10^1^ copies/μL. Finally, we performed a singleplex real-time PCR amplification of the four 10-fold-diluted plasmids as PCR templates to generate the standard curve.

### 2.5. Reaction Conditions of the Singleplex Real-Time PCR

The system of this singleplex real-time PCR was composed of 10 μL 2 × T5 Fast qPCR Mix, 10 μL of forward primer, 0.6 μL (10 μM) of each of the reverse primers, 0.2 μL (10 μM) of TaqMan probe, 2 μL of template, and the rest of the volume was nuclease-free water. The following program was used to perform the amplification on a Roche LightCycler^®^ 96 Instrument (Roche, Basel, Switzerland): 95 °C for 180 s; 40 cycles of 95 °C for 10 s; and 52 °C for 40 s (Ramp 2.2 °C/s). At the conclusion of each cycle, the fluorescence signal was automatically captured.

### 2.6. Optimization of Reaction Conditions for Multiplex Real-Time PCR

For multiplex real-time PCR reaction systems, 10 μL of 2 × T5 Fast qPCR Mix was mixed with four primers (0.6 μL), probe (0.6 μL), template (1 μL), and nuclease-free water in a reaction system with a final volume of 20 μL. In order to investigate the ideal concentrations of primers and probes for this multiplex method, the multiplex reaction conditions were optimized using primers and probes of varying concentrations (10 μM). The method’s final primer and probe concentrations ranged from 300 nM to 500 nM and 100 nM to 300 nM, respectively. As templates, 2.18 × 10^6^ copies/μL of plasmid standards were used (Table 2).

### 2.7. Specificity of Multiplex Real-Time RT-PCR

In order to assess the specificity of this multiplex RT-qPCR, we used standard DNA or cDNA of major porcine viruses as amplification templates, including CSFV, PRRSV, PRV, PCV2, and SVV. Meanwhile, the PEDV-M, TGEV-N, PDCoV-N, and PoRVA-VP7 recombinant plasmids were used as a positive control, and the nuclease-free water served as a negative template control.

### 2.8. Sensitivity of Multiplex Real-Time PCR

To assess the limit of detection (LOD) of the multiplex real-time PCR method, we performed a 10-fold serial dilution of the standard plasmid ranging from 2.18 × 10^7^ to 2.18 × 10^1^ copies/μL. Then, we amplified the diluted plasmid as a template using the multiplex method. Subsequently, to determine the reliable LOD of this method, we used the method to amplify 15 replicates of plasmids at concentrations from 2.18 × 10^3^ to 2.18 × 10^1^ copies/μL, and a detection rate ≥ 90% was determined to be actual and reliable.

### 2.9. Repeatability of Multiplex Real-Time PCR

To evaluate the reproducibility, ten-fold serially diluted standard plasmids between 2.18 × 10^7^ copies/μL to LOD were employed to investigate the multiplex real-time PCR coefficients of variation (CV%). In addition, for intra-assay repeatability, each sample was replicated three times. The assays were independently repeated three times using a different batch of standard plasmids for inter-assay repeatability.

### 2.10. Clinical Sample Detection

Using the above-developed multiplex real-time RT-PCR method, the four diarrhea viruses, PEDV, TGEV, PDCoV, and PoRVA, were detected in 97 samples stored in our laboratory between 2020 and 2023 for clinical delivery. These clinical samples consisted mainly of small intestine tissue and diarrheal feces and were treated in the same way as the RNA extraction procedure described above. To validate the detection accuracy, the results of this multiplex real-time RT-PCR on clinical samples were compared with the results of a conventional RT-PCR assay and confirmed by DNA sequencing.

## 3. Results

### 3.1. Establishment of Single Real-Time PCR Standard Curves

The recombinant plasmids, ranging from 2.18 × 10^8^ to 2.18 × 10^1^ copies/μL, were used to create standard curves. The standard curves showed a good amplification efficiency and correlation coefficient: PEDV (R^2^ = 0.9995; Efficiency = 101%), TGEV (R^2^ = 0.9998; Efficiency = 95%), PDCoV (R^2^ = 0.9950; Efficiency = 99%), and PoRVA (R^2^ = 0.9984; Efficiency = 110%); all standard curves were generated using GraphPad Prism 8.0 software. The results showed that our standard plasmids were reliable and that primers and probes were qualified (Figure 1).

### 3.2. Establishment and Optimization of the Multiplex Real-Time PCR Reaction Method

During the establishment and optimization of the multiplex real-time PCR method, due to mutual interference between different fluorophores, various concentrations of primers and probes can result in different amplification efficiencies. In this regard, we designed a combination of different concentrations of probes and primers, the probe ranging from 100 nM to 300 nM and the primer ranging from 300 nM to 500 nM, and then selected the optimal amplification curve. The results showed that the optimal concentration of probes and primers was a combination of 100 nM and 300 nM (Figure 2).

### 3.3. Specificity of the Multiplex Real-Time RT-PCR

In order to evaluate the specificity of this multiplex real-time RT-PCR method, six other major common viruses in swine were used as detection templates, including CSFV, PRRSV, SVV, PRV, and PCV2. Then, the recombinant plasmids of PEDV-M, TGEV-N, PDCoV-N, and PoRVA-VP7 were used as the positive control, and the concentration of the four plasmids was uniformly selected to be 2.18 × 10^3^ copies/μL. Meanwhile, nuclease-free water was used as the negative control for amplification. The results showed that only four positive amplification curves were presented, indicating that only all four target viruses in this study were detected (Figure 3), proving that the specificity of the multiplex method was reasonable.

### 3.4. Sensitivity of the Multiplex Real-Time PCR

To determine the sensitivity of this multiplex real-time PCR method, four recombinant plasmids were added to a reaction system after 10-fold serial dilution ranging from 2.18 × 10^7^ copies/μL to 2.18 × 10^1^ copies/μL, the results showed that the method for every viral pathogen was effectively established at the limit of detection (LOD) at 2.18 × 10^1^ copies/μL. However, a recombinant-plasmid-positive detection rate of 2.18 × 10^1^ copies/μL was found to be less than 90% in follow-up experiences, so the reliable LOD was judged to be 2.18 × 10^2^ copies/μL (Table 3, Figure 4).

### 3.5. Repeatability Test of the Multiplex Real-Time PCR

The four standard plasmids were used as templates for PCR amplification after 10-fold dilution, and both the intra-assay and the inter-assay were repeated three times; the results showed that the variation coefficients (CV%) of Cq values in the intra-assay and inter-assay tests ranged from 0.08% to 1.64% and 0.11% to 3.18%, respectively, indicating that this multiplex real-time PCR method is repeatable and stable (Table 4).

### 3.6. Detection of the Clinical Samples

The established multiplex real-time RT-PCR method was employed to determine the 97 clinical samples collected from Shandong Province. As shown in Figure 5, the results showed that positive rates of PEDV, TGEV, PDCoV, and PoRVA were 39.17% (38/97), 8.25% (8/97), 5.15% (5/97), and 26.80% (26/97), respectively. Moreover, the co-infection rates of PEDV + PoRVA, PEDV + TGEV, and PoRVA + TGEV were 12.37% (12/97), 5.15% (5/97), and 3.09% (3/97), respectively, while the PEDV + TGEV + PoRVA mixed infection rate was 3.09% (3/97). Subsequently, our method was compared with the results of a conventional single RT-PCR assay, the coincidence rate of the two methods was 100%. The five randomly selected samples that returned positive results were sequenced and identified with the corresponding viral gene fragments.

## 4. Discussion

China is a largely agricultural country, and porcine enterovirus outbreaks have severely affected the farm industry. The epidemics of four viruses, PEDV, TGEV, PDCoV, and PoRVA, have undermined the agricultural economy; they threaten public safety for both humans and animals due to their cross-species transmission and may be steadily adapting to a new host. For instance, the sudden outbreak of COVID-19, caused by Severe Acute Respiratory Syndrome Coronavirus Type 2 (SARS-CoV-2) virus, is estimated to have originated in bats and then spread to humans [25]. With the current rapid development of the farming industry, the link between humans and pigs is becoming closer, which further increases the possibility of cross-species transmission of viruses to humans.

Currently, accurate detection of animal pathogens is performed in the laboratory; there are a number of serological assays for these four diarrhea viruses, for example, researchers have developed multiplex ELISA methods that can be used to target PDCoV, TGEV, PEDV, and PoRV [26,27,28]. However, these four porcine enteroviruses predominantly infect suckling piglets, which have an immature immune system, so molecular diagnosis is relatively preferable to serologic diagnosis. Currently, singleplex and multiplex qPCRs have been widely used for the molecular detection of porcine diarrhea viruses. Still, there is no quadruplex real-time RT-PCR method that can simultaneously detect PEDV, TGEV, PDCoV, and RVA. For this reason, we have successfully developed a sensitive, specific, and cost-effective multiplex qPCR method.

In this study, the method requires the design of primer probes for the N protein gene of TGEV and PDCoV, the M protein gene of PEDV, and the VP7 protein gene of PoRVA, respectively, and the sequences of the selected genes are all highly conserved and not easily mutated, which allows the method to be used for a much more extended period. The test results also showed that the four pairs of probes and primers could only bind to their respective target sequences, and no cross-amplification between viruses or primer dimerization was found, which indicates that the specificity was excellent. However, bovine viral diarrhea virus (BVDV), which is currently increasingly prevalent in swine herds and, similarly, generates diarrhea symptoms in piglets [29,30,31], was not detected in the specificity test, which would have likely complicated the clinical differential diagnosis. In addition, theoretically, fluorescent probes can interfere with each other, thus affecting the final LOD, especially in the case of multiple pairs of probes. The results of the multiplex sensitivity test show that the effective LOD can reach 10^2^ copies/μL, indicating that the probes are designed to work well together. Finally, we speculate that the sensitivity of this study may be higher if the minor groove binder (MGB) probes are used for labeling, which can effectively reduce the fluorescence background signal.

To investigate the prevalence of PEDV, TGEV, PDCoV, and PoRVA in Shandong Province, China, from 2020 to 2023, we analyzed 97 clinical samples using multiplex real-time RT-PCR. The findings revealed that positive rates of PEDV, TGEV, PDCoV, and PoRVA were 39.17% (38/97), 8.25% (8/97), 5.15% (5/97), and 26.80% (26/97), respectively. Based on the results, it can be concluded that the positivity rates of PEDV and PoRVA dominate the porcine diarrhea viruses in Shandong Province. Meanwhile, a study by Li et al. showed a significant increase in PEDV positivity in Hunan and Hubei Provinces during 2020–2021 and a gradual increase in PDCoV and PoRVA positivity in 2020–2022 as the main pathogens responsible for porcine diarrhea in the region at that time; they also found that PEDV-PoRVA was the dominant co-infection mode [32]. Their results largely support our findings. However, in this study, we did not find any mixed infection with PDCoV in the co-infection assay and only found different degrees of mixed infection with the other three viruses. Based on the available epidemiologic findings, this may be due to the small number of clinical samples we tested. For example, Zhang et al. investigated 2987 diarrhea samples collected from 168 pig farms in five southern Chinese provinces between 2012 and 2018. The results showed that, among the 2987 samples, the separate infection rates of PDCoV, PoRVA, PEDV, and TGEV were 14.23% (425/2987), 0.60% (18/2987), 45.53% (1360/2987), and 0.33% (10/2987). They also reported that PEDV and PDCoV were the two viruses with the highest detection rates and that co-infections of these two viruses were also the most frequent, with a mean detection rate of 12.72% (380/2987) [21]. In addition, there are a number of related studies with similar findings that show the presence of PDCoV in co-infections [2,32,33,34,35]. They tested a large number of clinical samples, and the findings indicated the presence of PDCoV co-infection in all of them, which, to some extent, explains the absence of PDCoV in our study, and we will continue to conduct clinical tests to further refine the data in the next step.

In this study, a TaqMan-probe-based real-time RT-PCR method was developed to detect four pathogens in a single PCR reaction for more efficient detection of co-infections. The method has good specificity and sensitivity and is an effective tool for clinical differential diagnosis and epidemiologic investigation.

## Figures and Tables

**Figure 1 vetsci-11-00305-f001:**
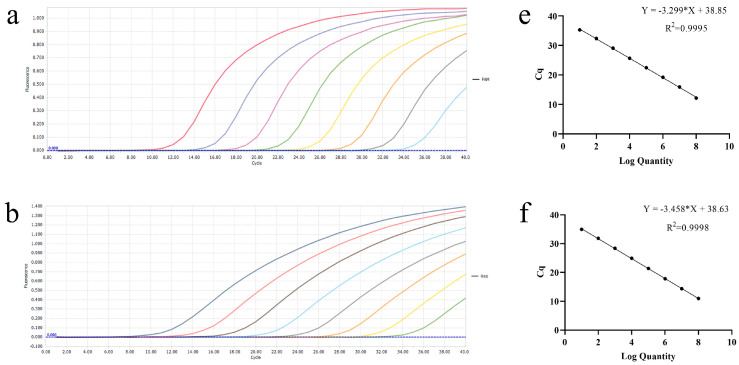
Establishment of amplification curves and standard curves. (**a**–**d**) Amplification curves of PEDV, TGEV, PDCoV, and PoRVA. (**e**–**h**) Standard curves of PEDV, TGEV, PDCoV, and PoRVA.

**Figure 2 vetsci-11-00305-f002:**
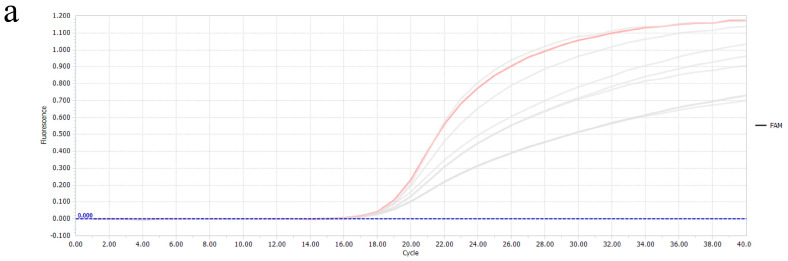
Probe and primer combinations at different concentrations. (**a**–**d**) Amplification curves of PEDV, TGEV, PDCoV, and PoRVA with different probe and primer concentrations. The pink lines are the optimal amplification curves, respectively.

**Figure 3 vetsci-11-00305-f003:**
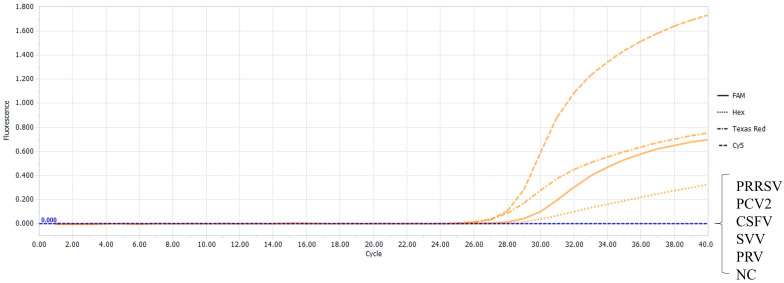
Specificity assays of the multiplex real-time RT-PCR. The four amplification curves represent the four positive controls for PEDV, TGEV, PDCoV, and PoRVA. No fluorescent signal was observed for other swine pathogen samples and the negative control (NC).

**Figure 4 vetsci-11-00305-f004:**
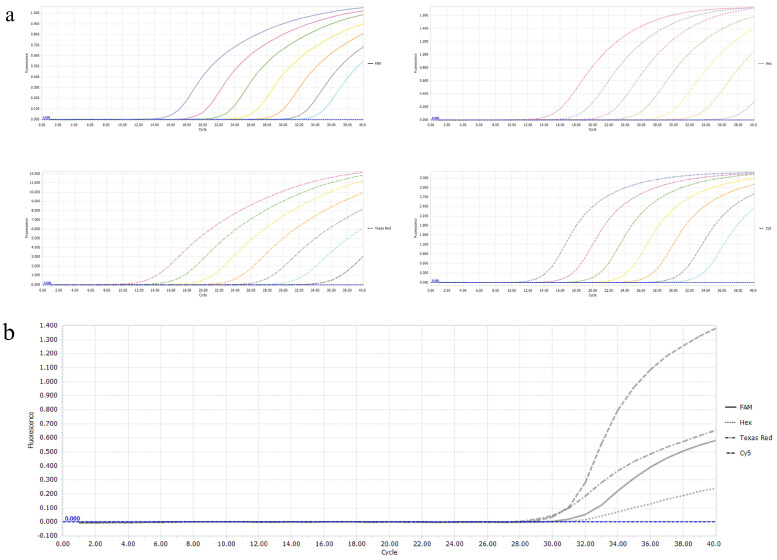
Sensitivity assay of the multiplex real−time RT−PCR. (**a**) Amplification curves are FAM−PEDV, Hex−TGEV, Texas Red−PDCoV, and Cy5−PoRVA, respectively. (**b**) Amplification curves were created by using the standard plasmid (2.18 × 10^2^ copies/μL) of PEDV, TGEV, PDCoV, and PoRVA.

**Figure 5 vetsci-11-00305-f005:**
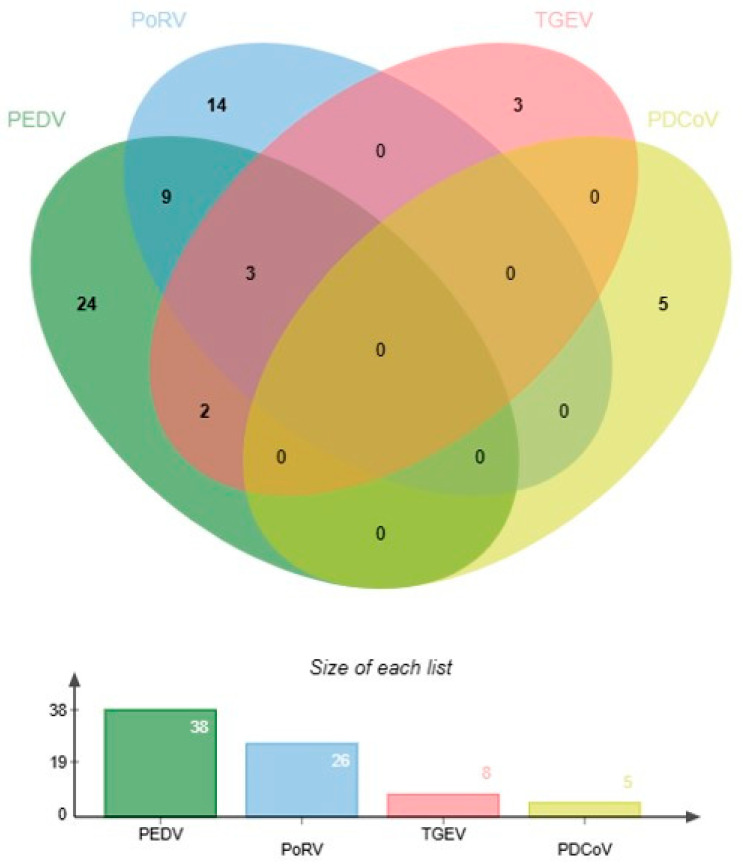
Results of multiplex real-time RT-PCR for clinical samples. This Venn diagram, plotted via the (bioinformatics.com.cn) website, shows individual infections and co-infections in the samples detected.

**Table 1 vetsci-11-00305-t001:** Primer and probe sequences.

Pathogens	Primers and Probe	Sequences (5′ to 3′)	Size (bp)	Gene
PEDV	Forward	TACTCTGCGTTCTTGTATGG	92	M
	Reverse	CTAGCCCATGCATCAAAAAG		
	Probe	TAMRA-AATAGCCATCTTGACACCATACA-FAM		
TGEV	Forward	GTGACAAGATTTTATGGAAC	157	N
	Reverse	CTTCCTTTGAAGTCCAATAG		
	Probe	HEX-CTAGACACAGATGGAACACATTCA-BHQ1		
PDCoV	Forward	ACTCCCTCCTAATGATAC	87	N
	Reverse	GCAACATGAGGTTTAATAG		
	Probe	Texas Red-CAGCAACCACTCGTGTTACTT-BHQ2		
PoRV-A	Forward	CGAACACATGTACTATAAGA	99	VP7
	Reverse	CAGCTGTTATGTCAAGTATA		
	Probe	Cy5-TTGGACCTCCTACCTGAATTAC-BHQ3		

**Table 2 vetsci-11-00305-t002:** The Cq values of PEDV, TGEV, PDCoV, and PoRVA detected by this multiplex real-time PCR assay with different probe and primer concentrations.

	PEDV				TGEV		
Probe concentration (nM)	Primer concentration (nM)			Probe concentration (nM)	Primer concentration (nM)		
	300	400	500		300	400	500
100	18.80	18.76	18.77	100	21.07	21.35	21.67
200	18.90	18.85	18.92	200	21.66	21.84	21.65
300	19.32	19.37	19.63	300	21.61	22.16	22.2
		PDCoV				PoRVA	
Probe concentration (nM)	Primer concentration (nM)			Probe concentration (nM)	Primer concentration (nM)		
	300	400	500		300	400	500
100	15.84	15.33	15.6	100	16.85	16.9	16.79
200	16.02	15.92	16.18	200	17.24	17.05	16.99
300	16.29	17.00	16.46	300	17.48	17.27	17.19

**Table 3 vetsci-11-00305-t003:** Sensitivity results of the multiplex real-time PCR.

Pathogens	Concentration	Total	Positive	Detection Rate	90% Detection Rate
PEDV	1000 copies/μL	15	15	100%	>90%
	100 copies/μL	15	15	100%	>90%
	10 copies/μL	15	9	60%	<90%
TGEV	1000 copies/μL	15	15	100%	>90%
	100 copies/μL	15	15	100%	>90%
	10 copies/μL	15	0	0%	<90%
PDCoV	1000 copies/μL	15	15	100%	>90%
	100 copies/μL	15	15	100%	>90%
	10 copies/μL	15	3	20%	<90%
PoRVA	1000 copies/μL	15	15	100%	>90%
	100 copies/μL	15	15	100%	>90%
	10 copies/μL	15	6	40%	<90%

Negative was determined by Cq ≥ 35 and positive was validated by Cq < 35.

**Table 4 vetsci-11-00305-t004:** Repeatability results (Cq value) of the multiplex real-time PCR.

Plasmid	Dilutions	Intra-Assay		Inter-Assay	
		Mean ± SD	CV%	Mean ± SD	CV%
PoRVA	10^7^	14.92 ± 0.05	0.37	15.00 ± 0.09	0.60
	10^6^	17.35 ± 0.01	0.10	17.38 ± 0.04	0.25
	10^5^	22.25 ± 0.13	0.59	20.27 ± 0.64	3.18
	10^4^	24.63 ± 0.08	0.34	24.70 ± 0.07	0.28
	10^3^	29.35 ± 0.34	1.17	29.31 ± 0.25	0.86
	10^2^	32.51 ± 0.53	1.64	32.95 ± 0.35	1.08
	NTC	ND	ND	ND	ND
TGEV	10^7^	15.27 ± 0.04	0.29	15.32 ± 0.07	0.46
	10^6^	19.11 ± 0.08	0.40	19.04 ± 0.13	0.66
	10^5^	21.11 ± 0.04	0.19	21.19 ± 0.05	0.23
	10^4^	24.14 ± 0.03	0.13	24.16 ± 0.03	0.12
	10^3^	27.58 ± 0.08	0.28	27.63 ± 0.06	0.21
	10^2^	31.67 ± 0.15	0.47	31.55 ± 0.36	1.15
	NTC	ND	ND	ND	ND
PEDV	10^7^	15.03 ± 0.06	0.40	15.04 ± 0.14	0.91
	10^6^	18.36 ± 0.02	0.08	18.40 ± 0.02	0.11
	10^5^	22.40 ± 0.11	0.50	23.34 ± 0.45	1.91
	10^4^	26.57 ± 0.07	0.27	26.74 ± 0.25	0.92
	10^3^	28.99 ± 0.25	0.87	28.88 ± 0.05	1.74
	10^2^	32.42 ± 0.33	1.03	32.55 ± 0.46	1.37
	NTC	ND	ND	ND	ND
PDCoV	10^7^	11.82 ± 0.15	1.27	12.44 ± 0.32	2.54
	10^6^	15.68 ± 0.08	0.48	15.74 ± 0.13	0.85
	10^5^	19.27 ± 0.08	0.42	19.13 ± 0.23	1.18
	10^4^	22.54 ± 0.08	0.35	22.61 ± 0.26	1.14
	10^3^	25.23 ± 0.12	0.46	25.44 ± 0.40	1.58
	10^2^	28.02 ± 0.14	0.5	28.50 ± 0.47	1.65
	NTC	ND	ND	ND	ND

## Data Availability

All data generated during the current study are included in the manuscript. Additional data related to this article may be requested from the corresponding author.

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
