# Peer review of "Development and Clinical Application of a Molecular Assay for Four Common Porcine Enteroviruses"

_vetsci, 2024, doi:10.3390/vetsci11070305_

Round 1
Reviewer 1 Report
Comments and Suggestions for Authors
Dear Authors!
You described in your manuscript a quadruplex PCR method to simultaneously detect PEDV, TGEV, PDCoV and rotavirus A. The material and methods and the results are well described and seem to be able to be easily repeated. However, I would recommend to explicitely talk of rotavirus A since the primers obviously only detect rotavirus A. I am also wondering how much of your text is generated by any AI??? The style used is very often a style generated by AI. If you really should have used AI, I would recommend to mention this in the Mat&Met section which AI was used and how it was used. This should be standard nowadays in case of any use of AI.
Please find all my comments and corrections directly in the attached PDF version.

Comments on the Quality of English LanguageThe quality of the English is very good, but seems to be not the common English style used in scientific papers.
Author Response
Comments 1: Please add Simple Summary part.
Response 1: A simple summary has been added to the manuscript.
Comments 2: About rotavirus A and some spelling issues that you mentioned in your manuscript.
Response 1: We agree with all your suggestions in the manuscript. In addition, I did not use any AI tools in the process of manuscript writing, because my English writing ability is not very good, I may not express myself well. thanks!

Reviewer 2 Report
Comments and Suggestions for Authors
The work written by Xin et al. describes the application of a molecular biology method that detects four pathogens of the pig's digestive system (via real-time with Taqman probes). The work is very interesting and well written, and the sections are well divided and developed. The article presents scientific rigor in all its sections. The results of the study have excellent applicability. I suggest acceptance of this work after minor revisions.
1) Title: The title is a bit long and complicated. “Molecular assay” may replace “TaqMan Probe-Based Multiplex Real-Time RT-PCR Method." "Enteroviruses" could also be confused with a genus of viruses.
2) Line 25: What do the authors mean by "clinical samples"? 3) "Subsequently, five randomly selected clinical samples that tested positive were sent for sequencing verification, and all turned out to be gene fragments of the corresponding viruses.". Please rephrase this sentence. 4) Lines 168–169: Also in this section, the authors should better specify the origin of these samples. Were they suspected of being positive for these viruses? Had they already been tested with other methods? 5) I advise the authors to check the references, as they are double-numbered.
Author Response
Comments 1: The title is a bit long and complicated. “Molecular assay” may replace “TaqMan Probe-Based Multiplex Real-Time RT-PCR Method."
Response: The title of the manuscript has been changed according to your suggestions.
Comments 2: Line 25: What do the authors mean by "clinical samples"?
Response: In the manuscript Materials and Methods there are references to the sources of the clinical samples. The clinical test samples were obtained from 97 clinical diarrhea samples sent from morbid pig farms in various regions of Shandong Province from 2020 to 2023, which mainly included small intestines, intestinal contents, and feces.
Comments 3: "Subsequently, five randomly selected clinical samples that tested positive were sent for sequencing verification, and all turned out to be gene fragments of the corresponding viruses.". Please rephrase this sentence
Response: ok!
Comments 4: Lines 168–169: Also in this section, the authors should better specify the origin of these samples
Response: In the manuscript Materials and Methods there are references to the sources of the clinical samples.
Comments 5: Were they suspected of being positive for these viruses? Had they already been tested with other methods?
Response: These samples were obtained from clinical diarrhoea samples from swine farms where severe diarrhoea occurred, and we suspected viral diarrhoea based on those clinical signs, so we developed this multiplex qPCR method for detection. Previously, we did not use other methods to test these samples, so in this study, we tested these samples using the new method we developed and then repeated the validation with traditional RT-PCR and DNA sequencing to ensure the accuracy of the results.
Comment 6: I advise the authors to check the references, as they are double-numbered.
Response: Revised

Reviewer 3 Report
Comments and Suggestions for Authors
In this manuscript, the authors established a multiplex RT-qPCR method for the detection of PEDV, TGEV, PDCoV and PoRV simultaneously. Although similar reports have been published before, the viruses detected are not exactly the same, so they still have clinical reference significance. There are only some minor typos that need to be corrected. For example, the r in Repeatability on line 24 should be lowercase.
Author Response
Comment 1: There are only some minor typos that need to be corrected. For example, the r in Repeatability on line 24 should be lowercase
Response:Thank you very much for your suggestions, which have been revised in the manuscript.
